# Active Learning with Oracle Epiphany

**Tzu-Kuo Huang** *
Uber Advanced Technologies Group
Pittsburgh, PA 15201

**Lihong Li**
Microsoft Research
Redmond, WA 98052

**Ara Vartanian**
University of Wisconsin–Madison
Madison, WI 53706

**Saleema Amershi**
Microsoft Research
Redmond, WA 98052

**Xiaojin Zhu**
University of Wisconsin–Madison
Madison, WI 53706

## Abstract

We present a theoretical analysis of active learning with more realistic interactions with human oracles. Previous empirical studies have shown oracles abstaining on difficult queries until accumulating enough information to make label decisions. We formalize this phenomenon with an "oracle epiphany model" and analyze active learning query complexity under such oracles for both the realizable and the agnostic cases. Our analysis shows that active learning is possible with oracle epiphany, but incurs an additional cost depending on when the epiphany happens. Our results suggest new, principled active learning approaches with realistic oracles.

## 1 Introduction

There is currently a wide gap between theory and practice of active learning with oracle interaction. Theoretical active learning assumes an omniscient oracle. Given a query $x$, the oracle simply answers its label $y$ by drawing from the conditional distribution $p(y \mid x)$. This oracle model is motivated largely by its convenience for analysis. However, there is mounting empirical evidence from psychology and human-computer interaction research that humans behave in far more complex ways. The oracle may abstain on some queries [Donmez and Carbonell, 2008] (note this is distinct from classifier abstention [Zhang and Chaudhuri, 2014, El-Yaniv and Wiener, 2010]), or their answers can be influenced by the identity and order of previous queries [Newell and Ruths, 2016, Sarkar et al., 2016, Kulesza et al., 2014] and by incentives [Shah and Zhou, 2015]. Theoretical active learning has yet to account for such richness in human behaviors, which are critical to designing principled algorithms to effectively learn from human annotators.

This paper takes a step toward bridging this gap. Specifically, we formalize and analyze the phenomenon of "oracle epiphany." Consider active learning from a human oracle to build a webpage classifier on *basketball sport vs. others*. It is well-known in practice that no matter how simple the task looks, the oracle can encounter difficult queries. The oracle may easily answer webpage queries that are obviously about basketball or obviously not about the sport, until she encounters a webpage on *basketball jerseys*. Here, the oracle cannot immediately decide how to label ("Does this jersey webpage qualify as a webpage about basketball?"). One solution is to allow the oracle to abstain by answering with a special I-don't-know label [Donmez and Carbonell, 2008]. More interestingly, Kulesza et al. [2014] demonstrated that with proper user interface support, the oracle may temporarily abstain on similar queries but then have an "epiphany": she may suddenly decide how to label all basketball apparel-related webpages. Empirical evidence in [Kulesza et al., 2014] suggests that epiphany may be induced by the accumulative effect of seeing multiple similar queries. If a future basketball-jersey webpage query arrives, the oracle will no longer abstain but will answer

with the label she determined during epiphany. In this way, the oracle improves herself on the subset of the input space that corresponds to basketball apparel-related webpages.

Empirical evidence also suggests that oracle abstention, and subsequent epiphany, may happen separately on different subsets of the input space. When building a *cooking vs. others* text classifier, Kulesza et al. [2014] observed oracle epiphany on a subset of cooking supplies documents, and separately on the subset of culinary service documents; on *gardening vs. others*, they observed separate oracle epiphany on plant information and on local garden documents; on *travel vs. others*, they observed separate oracle epiphany on photography, rental cars, and medical tourism documents.

Our contributions are three-fold: (*i*) We formalize oracle epiphany in Section 2; (*ii*) We analyze EPICAL, a variant of the CAL algorithm [Cohn et al., 1994], for realizable active learning with oracle epiphany in Section 3. (*iii*) We analyze Oracular-EPICAL, a variant of the Oracular-CAL algorithm [Hsu, 2010, Huang et al., 2015], for agnostic active learning in Section 4. Our query complexity bounds show that active learning is possible with oracle epiphany, although we may incur a penalty waiting for epiphany to happen. This is verified with simulations in Section 5, which highlights the nuanced dependency between query complexity and epiphany parameters.

## 2 Problem Setting

As in standard active learning, we are given a hypothesis class $\mathbb{H} \subseteq \mathbb{Y}^{\mathbb{X}}$ for some input space $\mathbb{X}$ and a binary label set $\mathbb{Y} \triangleq \{-1, 1\}$. There is an unknown distribution $\mu$ over $\mathbb{X} \times \mathbb{Y}$, from which examples are drawn IID. The marginal distribution over $\mathbb{X}$ is $\mu_{\mathbb{X}}$. Define the expected classification error, or risk, of a classifier $h \in \mathbb{H}$ to be $\text{err}(h) \triangleq \mathbf{E}_{(x,y) \sim \mu} [\mathbb{1}(h(x) \neq y)]$. As usual, the active learning goal is as follows: given any fixed $\epsilon, \delta \in (0, 1)$, we seek an active learning algorithm which, with probability at least $1 - \delta$, returns a hypothesis with classification error at most $\epsilon$ after sending a "small" number of queries to the oracle. What is unique here is an "oracle epiphany model."

The input space consists of two disjoint sets $\mathbb{X} = \mathbb{K} \cup \mathbb{U}$. The oracle knows the label for items in $\mathbb{K}$ (for "known") but initially does not know the labels in $\mathbb{U}$ (for "unknown"). The oracle will abstain if a query comes from $\mathbb{U}$ (unless epiphany happens, see below). Furthermore, $\mathbb{U}$ is partitioned into $K$ disjoint subsets $\mathbb{U} = \mathbb{U}^1 \cup \mathbb{U}^2 \cup \ldots \cup \mathbb{U}^K$. These correspond to the photograph/rental cars/medical tourism subsets in the *travel* task earlier. The active learner does not know the partitions nor $K$. When the active learner submits a query $x \in \mathbb{X}$ to the oracle, the learner will receive one of three outcomes in $\mathbb{Y}^+ \triangleq \{-1, 1, \bot\}$, where $\bot$ indicates I-don't-know abstention.

Importantly, we assume that epiphany is modeled as $K$ Markov chains: Whenever a unique $x \in \mathbb{U}^k$ is queried on some unknown region $k \in \{1, \ldots, K\}$ which did not experience epiphany yet, the oracle has a probability $\beta \in [0, 1]$ of epiphany on that region. If epiphany happens, the oracle then understands how to label everything in $\mathbb{U}^k$. In effect, the state of $\mathbb{U}^k$ is flipped from unknown to known. Epiphany is irrevocable: $\mathbb{U}^k$ will stay known from now on and the oracle will answer accordingly for all future $x$ therein. Thus the oracle will only answer $\bot$ if $\mathbb{U}^k$ remains unknown. The requirement for a *unique* $x$ is to prevent a trivial active learning algorithm which repeatedly queries the same $\bot$ item in an attempt to induce oracle epiphany. This requirement does not pose difficulty for analysis if $\mu_{\mathbb{X}}$ is continuous on $\mathbb{X}$, since all queries will be unique with probability one.

Therefore, our oracle epiphany model is parameterized by $(\beta, K, \mathbb{U}^1, \ldots, \mathbb{U}^K)$. All our analyses below will be based on this epiphany model. Of course, the model is only an approximation to real human oracle behaviors; In Section 6 we will discuss more sophisticated epiphany models for future work.

## 3 The Realizable Case

In this section, we study the realizable active learning case, where we assume there exists some $h^* \in \mathbb{H}$ such that the label of an example $x \in \mathbb{X}$ is $y = h^*(x)$. It follows that $\text{err}(h^*) = 0$. Although the realizability assumption is strong, the analysis is insightful on the role of epiphany. We will show that the worst-case query complexity has an additional $1/\beta$ dependence. We also discuss nice cases where this $1/\beta$ can be avoided depending on $\mathbb{U}$'s interaction with the disagreement region. Furthermore, our analysis focuses on the $K = 1$ case; that is, the oracle has only one unknown region $\mathbb{U} = \mathbb{U}^1$. This case is the simplest but captures the essence of the algorithm we propose in this section.

For convenience, we will drop the superscript and write $\mathbb{U}$. In the next section, we will eliminate both assumptions, and present and analyze an algorithm for the agnostic case with an arbitrary $K \geq 1$.

We modify the standard CAL algorithm [Cohn et al., 1994] to accommodate oracle epiphany. The modified algorithm, which we call EPICAL for "epiphany CAL," is given in Alg. 1. Like CAL, EPICAL receives a stream of unlabeled items; It maintains a version space; If the unlabeled item falls into the disagreement region of the version space the oracle is queried. The essential difference to CAL is that if the oracle answers $\perp$, no update to the version space happens. The stopping criterion ensures that the true risk of any hypothesis in the version space is at most $\epsilon$, with high probability.

---

**Algorithm 1** EPICAL

   Input: $\epsilon$, $\delta$, oracle, $\mathbb{X}$, $\mathbb{H}$
   Version space $\mathbb{V} \leftarrow \mathbb{H}$
   Disagreement region $\mathbb{D} \leftarrow \{x \in \mathbb{X} \mid \exists h, h' \in \mathbb{V}, h(x) \neq h'(x)\}$
   **for** $t = 1, 2, 3, \ldots$ **do**
      Sample an unlabeled example from the marginal distribution restricted to $\mathbb{D}$: $x_t \sim \mu_{\mathbb{X}|\mathbb{D}}$
      Query oracle with $x_t$ to get $y_t$
      **if** $y_t \neq \perp$ **then**
         $\mathbb{V} \leftarrow \{h \in \mathbb{V} \mid h(x_t) = y_t\}$
         $\mathbb{D} \leftarrow \{x \in \mathbb{X} \mid \exists h, h' \in \mathbb{V}, h(x) \neq h'(x)\}$
      **end if**
      **if** $\mu_{\mathbb{X}}(\mathbb{D}) \leq \epsilon$ **then**
         Return any $h \in \mathbb{V}$
      **end if**
   **end for**

---

Our analysis is based on the following observation: before oracle epiphany and ignoring all queries that result in $\perp$, EPICAL behaves exactly the same as CAL on an induced active-learning problem. The induced problem has input space $\mathbb{K}$, but with a projected hypothesis space we detail below. Hence, standard CAL analysis bounds the number of queries to find a good hypothesis in the induced problem. Now consider the sequence of probabilities of getting a $\perp$ label in each step of EPICAL. If these probabilities tend to be small, EPICAL will terminate with an $\epsilon$-risk hypothesis without even having to wait for epiphany. If these probabilities tend to be large, we may often hit the unknown region $\mathbb{U}$. But the number of such steps is bounded because epiphany will happen with high probability.

Formally, we define the induced active-learning problem as follows. The input space is $\bar{\mathbb{X}} \triangleq \mathbb{K}$, and the output space is still $\mathbb{Y}$. The sampling distribution is $\bar{\mu}_{\mathbb{X}}(x) \triangleq \mu_{\mathbb{X}}(x) \mathbb{1} (x \in \mathbb{K}) / \mu_{\mathbb{X}}(\mathbb{K})$. The hypothesis space is the *projection* of $\mathbb{H}$ onto $\bar{\mathbb{X}}$: $\bar{\mathbb{H}} \triangleq \{\bar{h} \in \mathbb{Y}^{\bar{\mathbb{X}}} \mid \exists h \in \mathbb{H}, \forall x \in \bar{\mathbb{X}} : \bar{h}(x) = h(x)\}$. Clearly, the induced problem is still realizable; let $\bar{h}^*$ be the projected target hypothesis. Let $\theta$ be the disagreement coefficient [Hanneke, 2014] for the original problem without unknown regions. The induced problem potentially has a different disagreement coefficient:

$$\bar{\theta} \triangleq \sup_{r > 0} \; r^{-1} \cdot \mathbf{E}_{x \sim \bar{\mu}_{\mathbb{X}}} \left[ \mathbb{1} \left( \exists \bar{h} \in \bar{\mathbb{H}} \text{ s.t. } \bar{h}^*(x) \neq \bar{h}(x), \mathbf{E}_{x' \sim \bar{\mu}_{\mathbb{X}}} \left[ \mathbb{1} \left( \bar{h}(x') \neq \bar{h}^*(x') \right) \right] \leq r \right) \right] .$$

Let $\bar{m}$ be the number of queries required for the CAL algorithm to find a hypothesis of $\epsilon/2$ risk with probability $1 - \delta/4$ in the induced problem. It is known [Hanneke, 2014, Theorem 5.1] that

$$\bar{m} \leq \bar{M} \triangleq \bar{\theta} \left( \dim(\bar{\mathbb{H}}) \ln \bar{\theta} + \ln \left( \frac{4}{\delta} \ln \frac{2}{\epsilon} \right) \right) \ln \frac{2}{\epsilon} .$$

where $\dim(\cdot)$ is the VC dimension. Similarly, let $m_{CAL}$ be the number of queries required for CAL to find a hypothesis of $\epsilon$ risk with probability $1 - \delta/4$ in the original problem, and we have $m_{CAL} \leq M_{CAL} \triangleq \theta \left( \dim(\mathbb{H}) \ln \theta + \ln \left( \frac{4}{\delta} \ln \frac{1}{\epsilon} \right) \right) \ln \frac{1}{\epsilon}$. Furthermore, define $m_{\perp} \triangleq |\{t \mid y_t = \perp\}|$ to be the number of queries in EPICAL for which the oracle returns $\perp$. We define $\mathbb{U}_t$ to be $\mathbb{U}$ for an iteration $t$ before epiphany, and $\emptyset$ after that. We define $\mathbb{D}_t$ to be the disagreement region $\mathbb{D}$ at iteration $t$. Finally, define the unknown fraction within disagreement as $\alpha_t \triangleq \mu_{\mathbb{X}}(\mathbb{D}_t \cap \mathbb{U}_t) / \mu_{\mathbb{X}}(\mathbb{D}_t)$. We are now ready to state the main result of this section.

**Theorem 1.** *Given any $\epsilon$ and $\delta$, EPICAL will, with probability at least $1 - \delta$, return an $\hat{h} \in \mathbb{H}$ with* $\text{err}(\hat{h}) \leq \epsilon$, *after making at most $M_{CAL} + \bar{M} + \frac{3}{\beta} \ln \frac{4}{\delta}$ queries.*

**Remark** The bound above consists of three terms. The first is the standard CAL query complexity bound with an omniscient oracle. The other two are the price we pay when the oracle is imperfect. The second term is the query complexity for finding a low-risk hypothesis in the induced active-learning problem. In situations where $\mu_{\mathbb{X}}(\mathbb{U}) = \epsilon/2$ and $\beta \ll 1$, it is hard to induce epiphany, but it suffices to find a hypothesis from $\bar{\mathbb{H}}$ with $\epsilon/2$ risk in the induced problem (which implies at most $\epsilon$ risk under the original distribution $\mu_{\mathbb{X}}$); it indicates $\bar{M}$ is unavoidable in some cases. The third term is roughly the extra query complexity required to induce epiphany. It is unavoidable in the worst case: when $\mathbb{U} = \mathbb{X}$, one has to wait for oracle epiphany to start collecting labeled examples to infer $h^*$; the average number of steps until epiphany is on the order of $1/\beta$. Finally, note that *not* all three terms contribute simultaneously to the query complexity of EPICAL. As we will see in the analysis and in the experiments, usually one or two of them will dominate, depending on how $\mathbb{U}$ interacts with the disagreement region. Summing them up simplifies our exposition, without changing the *order* of the worst-case bounds.

Our analysis starts with the definition of the following two events. Lemmas 2 and 3 show that they hold with high probability when running EPICAL; the proofs are delegated to Appendix A. Define:

$$\mathbf{E}_\perp \triangleq \left\{ m_\perp \leq \frac{1}{\beta} \ln \frac{4}{\delta} \right\} \quad \text{and} \quad \mathbf{E}_\alpha \triangleq \left\{ |\{t \mid \alpha_t > 1/2\}| \leq \frac{2}{\beta} \ln \frac{4}{\delta} \right\} .$$

**Lemma 2.** $\Pr\{\mathbf{E}_\perp\} \geq 1 - \delta/4$ .

**Lemma 3.** $\Pr\{\mathbf{E}_\alpha\} \geq 1 - \delta/4$.

**Lemma 4.** *Assume event $\mathbf{E}_\alpha$ holds. Then, the number of queries from $\mathbb{K}$ before oracle epiphany or before EPICAL terminates, whichever happens first, is at most $\bar{m} + \frac{2}{\beta} \ln \frac{4}{\delta}$.*

*Proof.* (sketch) Denote the quantity by $m$. Before epiphany, $\mathbb{V}$ and $\mathbb{D}$ in EPICAL behave in exactly the same way as in CAL on $\mathbb{K}$. It takes $\bar{m}$ queries to get to $\epsilon/2$ accuracy in $\mathbb{K}$ by the definition of $\bar{m}$. If $m \leq \bar{m}$, then $m < \bar{m} + \frac{2}{\beta} \ln \frac{4}{\delta}$ trivially, and we are done. Otherwise, it must be the case that $\alpha_t > 1/2$ for every step after $\mathbb{V}$ reaches $\epsilon/2$ accuracy on $\mathbb{K}$. Suppose not. Then there is a step $t$ where $\alpha_t \leq 1/2$. Note $\mathbb{V}$ reaching $\epsilon/2$ accuracy on $\mathbb{K}$ implies $\mu_{\mathbb{X}}(\mathbb{D}_t) - \mu_{\mathbb{X}}(\mathbb{D}_t \cap \mathbb{U}_t) \leq \epsilon/2$. Together with $\alpha_t = \mu_X(\mathbb{D}_t \cap \mathbb{U}_t)/\mu_X(\mathbb{D}_t) \leq 1/2$, we have $\mu_{\mathbb{X}}(\mathbb{D}_t) < \epsilon$. But this would have triggered termination of EPICAL at step $t$, a contradiction. Since we assume $\mathbf{E}_\alpha$ holds, we have $m \leq \bar{m} + \frac{2}{\beta} \ln \frac{4}{\delta}$. $\square$

*Proof of Theorem 1.* We will prove the query complexity bound, assuming (i) events $\mathbf{E}_\perp$ and $\mathbf{E}_\alpha$ hold; and (ii) $\bar{M}$ and $M_{CAL}$ successfully upper bound the corresponding query complexity of standard CAL. By Lemmas 2 and 3 and a union bound, the above holds with probability at least $1 - \delta$.

Suppose epiphany happens before EPICAL terminates. By event $\mathbf{E}_\perp$ and Lemma 4, the total number of queried examples before epiphany is at most $\bar{m} + \frac{3}{\beta} \ln \frac{4}{\delta}$. After epiphany, the total number of queries is no more than that of running CAL from scratch; this number is at most $M_{CAL}$. Therefore, the total query complexity is at most $\bar{M} + M_{CAL} + \frac{3}{\beta} \ln \frac{4}{\delta}$.

Suppose epiphany does *not* happen before EPICAL terminates. In this case, the number of queries in the unknown region is at most $\frac{1}{\beta} \ln \frac{4}{\delta}$ (event $\mathbf{E}_\perp$), and the number of queries in the known region is at most $\bar{m} + \frac{2}{\beta} \ln \frac{4}{\delta}$ (Lemma 4). Thus, the total number of queries is at most $\bar{M} + \frac{3}{\beta} \ln \frac{4}{\delta}$. $\square$

## 4 The Agnostic Case

In the agnostic setting the best hypothesis, $h^* \triangleq \arg\min_h \mathrm{err}(h)$, has a nonzero error. We want an active learning algorithm that, for a given accuracy $\epsilon > 0$, returns a hypothesis $h$ with small regret $\mathrm{reg}(h, h^*) \triangleq \mathrm{err}(h) - \mathrm{err}(h^*) \leq \epsilon$ while making a small number of queries. Among existing agnostic active learning algorithms we choose to adapt the Oracular-CAL algorithm, first proposed by Hsu [2010] and later improved by Huang et al. [2015]. Oracular-CAL makes no assumption on $\mathbb{H}$ or $\mu$, and can be implemented solely with an empirical risk minimization (ERM) subroutine, which is often well approximated by convex optimization over a surrogate loss in practice. This is a significant advantage over several existing agnostic algorithms, which either explicitly maintain a version space, as done in $A^2$ [Balcan et al., 2006], or require a *constrained* ERM routine [Dasgupta et al., 2007] that may not be well approximated efficiently in practice. IWAL [Beygelzimer et al., 2010] and Active

---

**Algorithm 2** Oracular-EPICAL

---

1: Set $c_1 \triangleq 4$ and $c_2 \triangleq 2\sqrt{6} + 9$. Let $\eta_0 \triangleq 1$ and $\eta_t \triangleq \frac{12}{t} \ln\left(\frac{32t|\mathbb{H}|\ln t}{\delta}\right), t \geq 1$.

2: Initialize labeled data $Z_0 \leftarrow \emptyset$, the version space $\mathbb{V}_1 \leftarrow \mathbb{H}$, and the ERM $h_1$ as any $h \in \mathbb{H}$.

3: **for** $t = 1, 2, \dots$ **do**

4:      Observe new example $x_t$, where $(x_t, y_t) \overset{i.i.d.}{\sim} \mu$.

5:      **if** $x_t \in \mathbb{D}_t \triangleq \{x \mid x \in \mathbb{X}, \exists (h, h') \in \mathbb{V}_t^2 \text{ s.t. } h(x) \neq h'(x)\}$ **then**

6:          Query oracle with $x_t$.

7:          $Z_t \leftarrow \begin{cases} Z_{t-1} \cup \{(x_t, y_t)\}, & \text{oracle returns } y_t. \\ Z_{t-1}, & \text{oracle returns } \perp. \end{cases}$

8:          $u_t \leftarrow \mathbb{1}\left(\text{oracle returns } \perp\right).$

9:      **else**

10:         $Z_t \leftarrow Z_{t-1} \cup \{(x_t, h_t(x_t))\}$. // update the labeled data with the current ERM's prediction

11:         $u_t \leftarrow 0$.

12:      **end if**

13:      $\text{err}(h, Z_t) \triangleq \frac{1}{t} \sum_{i=1}^{t} \mathbb{1}\left(x_i \in \mathbb{D}_i\right)(1-u_i)\mathbb{1}\left(h(x_i) \neq y_i\right) + \mathbb{1}\left(x_i \notin \mathbb{D}_i\right)\mathbb{1}\left(h(x_i) \neq h_i(x_i)\right).$

14:      $h_{t+1} \leftarrow \arg\min_{h \in \mathbb{H}} \text{err}(h, Z_t)$.

15:      $b_t \leftarrow \frac{1}{t} \sum_{i=1}^{t} u_i$.

16:      $\Delta_t \leftarrow c_1 \sqrt{\eta_t \text{err}(h_{t+1}, Z_t)} + c_2(\eta_t + b_t)$.

17:      $\mathbb{V}_{t+1} \leftarrow \{h \in \mathbb{H} \mid \text{err}(h, Z_t) - \text{err}(h_{t+1}, Z_t) \leq \Delta_t\}$.

18: **end for**

---

Cover [Huang et al., 2015] are agnostic algorithms that are implementable with an ERM routine, both using importance weights to correct for querying bias. But in the presence of $\perp$'s, choosing proper importance weights becomes challenging. Moreover, the improved Oracular-CAL [Huang et al., 2015] we use[2] has stronger guarantees than IWAL, and in fact, the best known worst-case guarantees among efficient, agnostic active learning algorithms.

Our proposed algorithm, Oracular-EPICAL, is given in Alg. 2. Note $t$ here counts unlabeled data, while in Alg. 1 it counts queries. Roughly speaking, Oracular-EPICAL also has an additive factor of $O(K/\beta)$ compared to Oracular-CAL's query complexity. It keeps a growing set $Z$ of labeled examples. If the unlabeled example $x_t$ falls in the disagreement region, the algorithm queries its label: when the oracle returns a label $y_t$, the algorithm adds $x_t$ and $y_t$ to $Z$; when the oracle returns $\perp$, no update to $Z$ happens. If $x_t$ is outside the disagreement region, the algorithm adds $x_t$ and the label predicted by the current ERM hypothesis $h_t(x_t)$ to $Z$. Alg. 2 keeps an indicator $u_t$, which records whether $\perp$ was returned on $x_t$, and it always updates the ERM and the version space after every new $x_t$. For simplicity we assume a finite $\mathbb{H}$; this can be extended to $\mathbb{H}$ with finite VC dimension.

The critical modification we make here to accommodate oracle abstention is that the threshold $\Delta_t$ defining the version space additively depends on the average number of $\perp$'s received up to round $t$. This allows us to show that Oracular-EPICAL retains the *favorable bias* guarantee of Oracular-CAL: with high probability, *all of the imputed labels are consistent with the classifications of $h^*$*, so imputation never pushes the algorithm away from $h^*$. Oracular-EPICAL only uses the version space in the disagreement test. With the same technique used by Oracular-CAL, summarized in Appendix B, the algorithm is able to perform the test solely with an ERM routine.

We now state Oracular-EPICAL's general theoretical guarantees, which hold for any oracle model, and then specialize them for the epiphany model in Section 2. We start with a consistency result:

**Theorem 5** (Consistency Guarantee). *Pick any $0 < \delta < 1/e$ and let $\Delta_t^* := c_1 \sqrt{\eta_t \text{err}(h^*)} + c_2(\eta_t + b_t)$. With probability at least $1 - \delta$, the following holds for all $t \geq 1$,*

$$\text{err}(h) - \text{err}(h^*) \leq 4\Delta_t^* \quad \textit{for all } h \in \mathbb{V}_{t+1}, \quad \textit{and} \tag{1}$$

$$\text{err}(h^*, Z_t) - \text{err}(h_{t+1}, Z_t) \leq \Delta_t. \tag{2}$$

All hypotheses in the current version space, including the current ERM, have controlled expected regrets. Compared with Oracular-CAL's consistency guarantee, this is worse by an additive factor of $O(b_t)$, the average number of $\perp$'s over $t$ examples. Importantly, $h^*$ always remains in the version space, as implied by (2). This guarantees that all predicted labels used by the algorithm are consistent with $h^*$, since the entire version space makes the same prediction. The query complexity bound is:

**Theorem 6** (Query Complexity Bound). *Let $Q_t \triangleq \sum_{i=1}^{t} \mathbb{1}(x_i \in \mathbb{D}_i)$ denote the total number of queries Alg. 2 makes after observing $t$ examples. Under the conditions of Theorem 5, with probability at least $1 - \delta$ the following holds: $\forall t > 0$, $Q_t$ is bounded by*

$$4\theta \operatorname{err}(h^*)t + \theta \cdot O\left( \sqrt{t \operatorname{err}(h^*) \ln(t|\mathbb{H}|/\delta) \ln^2 t} + \ln(t|\mathbb{H}|/\delta) \ln t + tb_t \ln t + 8 \ln(8t^2 \ln t/\delta) \right),$$

*where $\theta$ denotes the disagreement coefficient [Hanneke, 2014].*

Again, this result is worse than Oracular-CAL's query complexity [Huang et al., 2015] by an additive factor. The magnitude of this factor is less trivial than it seems: since the algorithm increases the threshold by $b_t$, it includes more hypotheses in the version space, which may cause the algorithm to query a lot more. However, our analysis shows that the number of queries only increases by $O(tb_t \ln t)$, i.e., $\ln t$ times the total number of $\perp$'s received over $t$ examples.

The full proofs of both theorems are in Appendix C. Here we provide the key ingredient. Consider an imaginary dataset $Z_t^\dagger$ where all the labels queried by the algorithm but not returned by the oracle are imputed, and define the error on this imputed data:

$$\operatorname{err}(h, Z_t^\dagger) \triangleq \frac{1}{t} \sum_{i=1}^{t} \mathbb{1}(x_i \in \mathbb{D}_i)\mathbb{1}(h(x_i) \neq y_i) + \mathbb{1}(x_i \notin \mathbb{D}_i)\mathbb{1}(h(x_i) \neq h_i(x_i)). \tag{3}$$

Note that the version space $\mathbb{V}_t$ and therefore the disagreement region $\mathbb{D}_t$ are still defined in terms of $\operatorname{err}(h, Z_t)$, not $\operatorname{err}(h, Z_t^\dagger)$. Also define the empirical regrets between two hypotheses $h$ and $h'$: $\operatorname{reg}(h, h', Z_t) \triangleq \operatorname{err}(h, Z_t) - \operatorname{err}(h', Z_t)$ and $\operatorname{reg}(h, h', Z_t^\dagger)$ on $Z_t^\dagger$ in the same way. The empirical error and regret on $Z_t^\dagger$ are not observable, but can be easily bounded by observable quantities:

$$\operatorname{err}(h, Z_t) \leq \operatorname{err}(h, Z_t^\dagger) \leq \operatorname{err}(h, Z_t) + b_t, \tag{4}$$

$$|\operatorname{reg}(h, h', Z_t) - \operatorname{reg}(h, h', Z_t^\dagger)| \leq b_t, \tag{5}$$

where $b_t = \sum_{i=1}^{t} u_i/t$ is also observable. Using a martingale analysis resembling Huang et al. [2015]'s for Oracular-CAL, we prove concentration of the empirical regret $\operatorname{reg}(h, h^*, Z_t^\dagger)$ to its expectation. For every $h \in \mathbb{V}_{t+1}$, the algorithm controls its empirical regret on $Z_t$, which bounds $\operatorname{reg}(h, h^*, Z_t^\dagger)$ by the above. This leads to a bound on the expected regret of $h$. The query complexity analysis follows the standard framework of Hsu [2010] and Huang et al. [2015].

Next, we specialize the guarantees to the oracle epiphany model in Section 2:

**Corollary 7.** *Assume the epiphany model in Section 2. Fix $\epsilon > 0, \delta > 0$. Let $\tilde{d} \triangleq \ln(|\mathbb{H}|/(\epsilon\delta))$, $\widetilde{K} \triangleq K\ln(K/\delta)$ and $e^* \triangleq \operatorname{err}(h^*)$. With probability at least $1 - \delta$, the following holds: The ERM hypothesis $h_{t_\epsilon+1}$ satisfies $\operatorname{err}(h_{t_\epsilon+1}) - e^* \leq \epsilon$, where $t_\epsilon = O\left( \frac{\tilde{d}e^*}{\epsilon^2} + \frac{1}{\epsilon}\left(\tilde{d} + \frac{\widetilde{K}}{\beta}\right) \right)$, and the total number of queries made up to round $t_\epsilon$ is*

$$\theta \cdot O\left( \frac{e^*}{\epsilon}\left(\frac{\tilde{d}\cdot e^*}{\epsilon} + \frac{\widetilde{K}}{\beta}\right) + \ln\left(\left(\frac{e^*}{\epsilon^2} + \frac{1}{\epsilon}\right)\tilde{d} + \frac{\widetilde{K}}{\epsilon\beta}\right) \cdot \left(\left(\frac{e^*}{\epsilon} + 1\right)\tilde{d} + \frac{\widetilde{K}}{\beta}\right) \right).$$

The proof is in Appendix D. This corollary reveals how the epiphany parameters $K$ and $\beta$ affect query complexity. Setting $\widetilde{K} = 0$ recovers the result for a perfect oracle, showing that the (unlabeled) sample complexity $t_\epsilon$ worsens by an additive factor of $\widetilde{K}/(\beta\epsilon)$ in both realizable and agnostic settings. For query complexity, in the realizable setting the bound becomes $\theta \cdot O\left( \ln\left((\tilde{d} + \widetilde{K}/\beta)/\epsilon\right)(\tilde{d} + \widetilde{K}/\beta) \right)$. In the agnostic setting, the leading term in our bound is $\theta \cdot O\left((e^*/\epsilon)^2\tilde{d} + (\widetilde{K}e^*)/(\beta\epsilon)\right)$. In both cases, our bounds are worse by roughly an additive factor of $O(\widetilde{K}/\beta)$ than bounds for perfect oracles.

As for the effect of $\mathbb{U}$, the above corollary is a worst-case result: it uses an upper bound on $tb_t$ that holds even for $\mathbb{U} = \mathbb{X}$. For certain $U$'s the upper bound can be much tighter. For example, if $\mathbb{U} \cap \mathbb{D}_t = \emptyset$ for sufficiently large $t$, then $tb_t$ will be $O(1)$ for all $\beta$, with or without epiphany.

# 5 Experiments

To complement our theoretical results, we present two simulated experiments on active learning with oracle epiphany: learning a 1D threshold classifier and handwritten digit recognition (OCR). Specifically, we will highlight query complexity dependency on the epiphany parameter $\beta$ and on $\mathbb{U}$.

**EPICAL on 1D Threshold Classifiers.** Take $\mu_{\mathbb{X}}$ to be the uniform distribution over the interval $\mathbb{X} = [0, 1]$. Our hypothesis space is the set of threshold classifiers $\mathbb{H} = \{h_a : a \in [0, 1]\}$ where $h_a(x) = \mathbb{1}(x \geq a)$. We choose $h^* = h_{\frac{1}{2}}$ and set the target classification error at $\epsilon = 0.05$.

We illustrate epiphany with a single unknown region $K = 1, \mathbb{U} = \mathbb{U}^1$. However, we contrast two shapes of $\mathbb{U}$: in one set of experiments we set $\mathbb{U} = [0.4, 0.6]$ which contains the decision boundary 0.5. In this case, the active learner EPICAL must induce oracle epiphany in order to achieve $\epsilon$ risk. In another set of experiments $\mathbb{U} = [0.7, 0.9]$, where we expect the learner to be able to "bypass" the need for epiphany. Intuitively, this latter $\mathbb{U}$ could soon be excluded from the disagreement region. For both $\mathbb{U}$, we systematically vary the oracle epiphany parameter $\beta \in \{2^{-6}, 2^{-5}, \ldots, 2^0\}$. A small $\beta$ means epiphany is less likely per query, thus we expect the learner to spend more queries trying to induce epiphany in the case of $\mathbb{U} = [0.4, 0.6]$. In contrast, $\beta$ may not matter much in the case of $\mathbb{U} = [0.7, 0.9]$ since epiphany may not be required. Note that $\beta = 2^0 = 1$ reverts back to the standard active learning oracle, since epiphany always happens immediately. We run each combination of $\beta, \mathbb{U}$

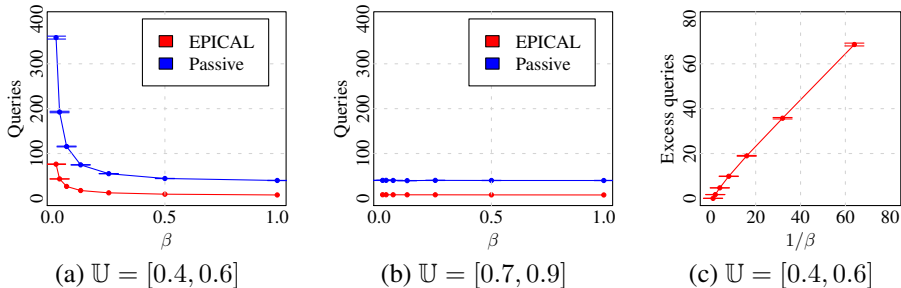

(a) $\mathbb{U} = [0.4, 0.6]$         (b) $\mathbb{U} = [0.7, 0.9]$         (c) $\mathbb{U} = [0.4, 0.6]$

Figure 1: EPICAL results on 1D threshold classifiers

for $10,000$ trials. The results are shown in Figure 1. As expected, (a) shows a clear dependency on $\beta$. This indicates that epiphany is necessary in the case $\mathbb{U} = [0.4, 0.6]$ for learning to be successful. In contrast, the dependence on $\beta$ vanishes in (b) when $\mathbb{U}$ is shifted sufficiently away from the target threshold (and thus from later disagreement regions). The oracle need not reach epiphany for learning to happen. Note (b) does not contradict with EPICAL query complexity analysis since Theorem 1 is a worst case bound that must hold true for all $\mathbb{U}$.

To further clarify the role of $\beta$, note EPICAL query complexity bound predicts an additive term of $O(1/\beta)$ on top of the standard CAL query complexities (i.e., both $\bar{M}$ and $M_{CAL}$). This term represents "excess queries" needed to induce epiphany. In Figure 1(c) we plot this excess against $\frac{1}{\beta}$ for $\mathbb{U} = [0.4, 0.6]$. Excess is computed as the number of EPICAL queries minus the average number of queries for $\beta = 1$. Indeed, we see a near linear relationship between excess queries and $1/\beta$.

Finally, as a baseline we compare EPICAL to passive learning. In passive learning $x_1, x_2, \ldots$ are chosen randomly according to $\mu_{\mathbb{X}}$ instead of adaptively. Note passive learning here is also subject to oracle epiphany. That is, the labels $y_t$ are produced by the same oracle epiphany model, some of them can be $\perp$ initially. Our passive learning simply maintains a version space. If it encounters $\perp$ it does not update the version space. All EPICAL results are better than passive learning.

**Oracular-EPICAL on OCR.** We consider the binary classification task of *5 vs. other digits* on MNIST [LeCun et al., 1998]. This allows us to design the unknown regions $\{\mathbb{U}^k\}$ as certain other digits, making the experiments more interpretable. Furthermore, we can control how confusable the $\mathbb{U}$ digits are to "5" to observe the influence on oracle epiphany.

Although Alg. 2 is efficiently implementable with an ERM routine, it still requires two calls to a supervised learning algorithm on every new example. To scale it up, we implement an approximate version of Alg. 2 that uses *online optimization* in place of the ERM. More details are in Appendix E. While being efficient in practice, this online algorithm may not retain Alg. 2's theoretical guarantees.

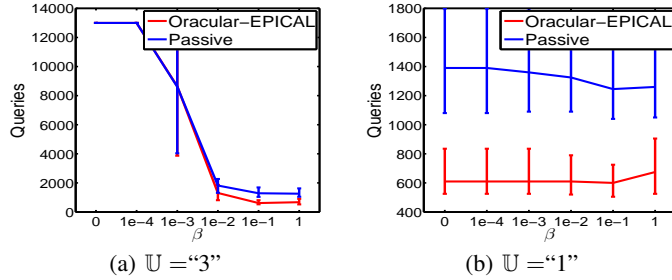

(a) $\mathbb{U} =$ "3"            (b) $\mathbb{U} =$ "1"

Figure 2: Oracular-EPICAL results on OCR.

We use epiphany parameters $\beta \in \{1, 10^{-1}, 10^{-2}, 10^{-3}, 10^{-4}, 0\}$, $K = 1$, and $\mathbb{U}$ is either "3" or "1". By using $\beta = 1$ and $\beta = 0$, we include the boundary cases where the oracle is perfect or never has an epiphany. The two different $\mathbb{U}$'s correspond to two contrasting scenarios: "3" is among the "nearest" digits to "5" as measured by the binary classification error between "5" and every other single digit, while "1" is the farthest. The two $\mathbb{U}$'s are about the same size, each covering roughly $10\%$ of the data. More details and experimental results with other choices of $\mathbb{U}$ can be found in Appendix E. For each combination of $\beta$ and $\mathbb{U}$, we perform 100 random trials. In each trial, we run both the online version of Alg. 2 and online passive logistic regression (also subject to oracle epiphany) over a randomly permuted training set of $60,000$ examples, and check the error of the online ERM on the $10,000$ testing examples every 10 queries from 200 up to our query budget of $13,000$. In each trial we record the smallest number of queries for achieving a test error of $4\%$. Fig. 2(a) and Fig. 2(b) show the median of this number over the 100 random trials, with error bars being the 25th and 75th quantiles. The effect of $\beta$ on query complexity is dramatic for the near $\mathbb{U} =$ "3" but subdued for the far $\mathbb{U} =$ "1". In particular, for $\mathbb{U} =$ "3" small $\beta$'s force active learning to query as many labels as passive learning. The flattening at $13,000$ at the end means no algorithm could achieve a $4\%$ test error within our query budget. For $\mathbb{U} =$ "1", active learning is always much better than passive regardless of $\beta$. Again, this illustrates that both $\beta$ and $\mathbb{U}$ affect the query complexity. As performance references, passive learning on the entire labeled training data achieves a test error of $2.6\%$, while predicting the majority class (non-5) has a test error of $8.9\%$.

## 6 Discussions

Our analysis reveals a worst case $O(1/\beta)$ term in query complexity due to the wait for epiphany, and we hypothesize $\Omega(K/\beta)$ to be the tight lower bound. This immediately raises the question: can we decouple active learning queries from epiphany induction? What if the learner can quickly induce epiphany by showing the oracle a screenful of unlabeled items at a time, without the oracle labeling them? This possibility is hinted in empirical studies. For example, Kulesza et al. [2014] observed epiphanies resulting from seeing items. Then there is a tradeoff between two learner actions toward the oracle: asking a query (getting a label or small contribution toward epiphany), or showing several items (not getting labels but potentially large contribution toward epiphany). One must formalize the cost and benefit of this tradeoff. Of course, real human behaviors are even richer. Epiphanies may be reversible on certain queries, where the oracle begins to have doubts on her previous labeling. Extending our model under more relaxed assumptions is an interesting open question for future research.

### Acknowledgments

This work is supported in part by NSF grants IIS-0953219, IIS-1623605, DGE-1545481, CCF-1423237, and by the University of Wisconsin-Madison Graduate School with funding from the Wisconsin Alumni Research Foundation.

## Footnotes

*Part of this work was done while the author was with Microsoft Research.

[2]This improved version of Oracular-CAL defines the version space using a tighter threshold than the one used by Hsu [2010], and has the same worst-case guarantees as Active Cover [Huang et al., 2015].

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
