[Supplementary Material]

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

## A    Additional Proofs for Section 3

*Proof of Lemma 2.* Suppose the oracle returns $\perp$ exactly $m$ times. It implies the oracle has no epiphany for at least the first $m$ steps. The probability of such an event is $(1-\beta)^m$. Let the right-hand side be $\delta/4$ and solve for $m$, we obtain

$$m = \frac{\ln(\delta/4)}{\ln(1-\beta)} \leq \frac{1}{\beta} \ln \frac{4}{\delta} .$$

$\square$

*Proof of Lemma 3.* Note that $\alpha_t > 1/2$ means in step $t$ the active learner is likely to propose a query $x_t$ that falls in $\mathbb{U}_t \subseteq \mathbb{U}$. Specifically, the probability that epiphany happens in step $t$, denoted by $\beta_t$, is given by

$$\beta_t = \beta \cdot \mu_{\mathbb{X}|\mathbb{D}_t}(x_t \in \mathbb{U}_t) + 0 \cdot \mu_{\mathbb{X}|\mathbb{D}_t}(x_t \notin \mathbb{U}_t) = \beta\alpha_t > \frac{\beta}{2} ,$$

where $\mathbb{D}_t$ is the disagreement region in step $t$ when $x_t$ is sampled. Suppose there are exactly $m$ such steps with $\alpha_t$; denote these steps by $t_1, t_2, \ldots, t_m$. It implies the oracle has no epiphany for at least the first $m$ steps, the probability of which is

$$\prod_{i=1}^{m}(1 - \beta_{t_i}) \leq \left(1 - \frac{\beta}{2}\right)^m .$$

Let the right-hand side be $\delta/4$ and solve for $m$, we obtain

$$m = \frac{\log(\delta/4)}{\log(1-\beta/2)} \leq \frac{2}{\beta} \log \frac{4}{\delta} .$$

$\square$

## B    Implementation of Oracular-EPICAL's Disagreement Test with ERM

Because Oracular-EPICAL's version space is defined in terms of empirical error, we are able to carry out the disagreement test using the following technique inspired by Dasgupta et al. [2007], which only relies on an ERM subroutine: To test whether $x \in \mathbb{D}_t$, we call the ERM subroutine to find

$$h' := \arg\min_{h\in\mathbb{H}} \quad \mathrm{err}(h, Z_{t-1}) + \Delta_{t-1}\mathbb{1}\left(h(x) \neq -h_t(x)\right),$$

where $h_t = \arg\min_{h\in\mathbb{H}} \mathrm{err}(h, Z_{t-1})$. In practice, this means we create a labeled example $(x, -h_t(x))$ with *a weight of* $(t-1)\Delta_{t-1}$, add it to $Z_{t-1}$ and feed the augmented data to a supervised learning algorithm, whose output will be $h'$. Then we return $\mathbb{1}\left(h'(x) = h_t(x)\right)$ as $\mathbb{1}\left(x \notin \mathbb{D}_t\right)$.

To see why this is a valid test, first consider $h'(x) \neq h_t(x)$. Since $h'$ minimizes the augmented empirical error, it is true that $\mathrm{err}(h', Z_{t-1}) \leq \mathrm{err}(h_t, Z_{t-1}) + \Delta_{t-1}$, implying $h' \in \mathbb{V}_t$ and therefore $x_t \in \mathbb{D}_t$. Now suppose $h'(x) = h_t(x)$. For all $h \in \mathbb{H}$ such that $h(x) = -h_t(x)$, it must be the case that $\mathrm{err}(h, Z_{t-1}) \geq \mathrm{err}(h', Z_{t-1}) + \Delta_{t-1} \geq \mathrm{err}(h_t, Z_{t-1}) + \Delta_{t-1}$, i.e, $h \notin \mathbb{V}_t$. This implies that $\forall h \in V_t, h(x) = h_t(x)$, so $x \notin \mathbb{D}_t$.

## C    General Analysis for Oracular-EPICAL

To analyze Alg. 2, we need some more notations. Let $\mathrm{reg}(h, h') \triangleq \mathrm{err}(h) - \mathrm{err}(h')$ denote the regret between two hypotheses $h$ and $h'$. Consider an imaginary dataset $Z_t^{\dagger}$ where all the labels queried by the algorithm but not returned by the oracle are imputed, and define the error on this imputed data:

$$\mathrm{err}(h, Z_t^{\dagger}) \triangleq \frac{1}{t}\sum_{i=1}^{t} \mathbb{1}\left(x_i \in \mathbb{D}_i\right)\mathbb{1}\left(h(x_i) \neq y_i\right) + \mathbb{1}\left(x_i \notin \mathbb{D}_i\right)\mathbb{1}\left(h(x_i) \neq h_i(x_i)\right). \quad (6)$$

Note that the version space $\mathbb{V}_t$ and therefore the disagreement region $\mathbb{D}_t$ are still defined in terms of $\mathrm{err}(h, Z_t)$, not $\mathrm{err}(h, Z_t^{\dagger})$. Also define the empirical regrets between two hypotheses $h$ and $h'$:

$$\mathrm{reg}(h, h', Z_t) = \mathrm{err}(h, Z_t) - \mathrm{err}(h', Z_t), \quad (7)$$

$$\mathrm{reg}(h, h', Z_t^{\dagger}) = \mathrm{err}(h, Z_t^{\dagger}) - \mathrm{err}(h', Z_t^{\dagger}). \quad (8)$$

The two quantities (6) and (8) are not observable, but can be easily bounded by observable quantities:

$$\mathrm{err}(h, Z_t) \leq \mathrm{err}(h, Z_t^\dagger) \leq \mathrm{err}(h, Z_t) + b_t, \tag{9}$$

$$|\mathrm{reg}(h, h', Z_t) - \mathrm{reg}(h, h', Z_t^\dagger)| \leq b_t, \tag{10}$$

where $b_t = \frac{1}{t} \sum_{i=1}^{t} u_i$ is also observable. In addition to empirical quantities, we also need their expectations conditioned on all history. More formally, let $\mathcal{F}_t \triangleq \sigma(\{(x_i, y_i, u_i)\}_{i=1}^{t})$ denote the $\sigma$-algebra on all history up to round $t$, and let $\mathbf{E}_t[\cdot] \triangleq \mathbf{E}[\cdot \mid \mathcal{F}_{t-1}]$ denote expectation at round $t$ conditioned on all history up to round $t - 1$.

**Expected error and regret** Define the following expected error and regret terms at round $t$:

$$\mathrm{err}_t^\dagger(h) \triangleq \mathbf{E}_t\left[\mathbb{1}\left(x_t \in \mathbb{D}_t\right) \mathbb{1}\left(h(x_t) \neq y_t\right) + \mathbb{1}\left(x_t \notin \mathbb{D}_t\right) \mathbb{1}\left(h(x_t) \neq h_t(x_t)\right)\right], \tag{11}$$

$$\mathrm{reg}_t^\dagger(h, h') \triangleq \mathrm{err}_t^\dagger(h) - \mathrm{err}_t^\dagger(h') \tag{12}$$

and their averages

$$\widetilde{\mathrm{err}}_t(h) \triangleq \frac{1}{t} \sum_{i=1}^{t} \mathrm{err}_i^\dagger(h), \qquad \widetilde{\mathrm{reg}}_t(h, h') \triangleq \frac{1}{t} \sum_{i=1}^{t} \mathrm{reg}_i^\dagger(h). \tag{13}$$

Also define the following expected error and regret terms restricted to disagreement regions.

$$\mathrm{err}_t(h) := \mathbf{E}_{(x,y) \sim \mu}\left[\mathbb{1}\left(x \in \mathbb{D}_t\right) \mathbb{1}\left(h(x) \neq y\right)\right], \tag{14}$$

$$\overline{\mathrm{err}}_t(h) := \frac{1}{t} \sum_{i=1}^{t} \mathrm{err}_i(h), \tag{15}$$

$$\mathrm{reg}_t(h) := \mathrm{err}_t(h) - \mathrm{err}_t(h^*), \tag{16}$$

$$\overline{\mathrm{reg}}_t(h) := \overline{\mathrm{err}}_t(h) - \overline{\mathrm{err}}_t(h^*). \tag{17}$$

We start with two important lemmas.

**Lemma 8** (Favorable Bias). $\forall i \geq 1, \forall h \in \mathbb{H}, \forall \bar{h} \in \mathbb{V}_i,$ *we have*

$$\mathrm{reg}_i^\dagger(h, \bar{h}) \geq \mathrm{reg}(h, \bar{h}). \tag{18}$$

*Proof.* Pick any $i \geq 1, h \in \mathbb{H}$ and $\bar{h} \in \mathbb{V}_i$. Note that the definitions of $\mathrm{reg}_i^\dagger(h, \bar{h})$ and $\mathrm{reg}(h, \bar{h})$ only differ on $x \notin \mathbb{D}_i$, and $\forall x \notin \mathbb{D}_i$, $\bar{h}(x) = h_i(x)$. We thus have

$$\mathrm{reg}_i^\dagger(h, \bar{h}) - \mathrm{reg}(h, \bar{h})$$

$$= \mathbf{E}_{(x,y) \sim \mu}\left[\mathbb{1}\left(x \notin \mathbb{D}_i\right)\left(\left(\mathbb{1}\left(h(x) \neq h_i(x)\right) - \mathbb{1}\left(\bar{h}(x) \neq h_i(x)\right)\right) - \left(\mathbb{1}\left(h(x) \neq y\right) - \mathbb{1}\left(\bar{h}(x) \neq y\right)\right)\right)\right]$$

$$= \mathbf{E}_{(x,y) \sim \mu}\left[\mathbb{1}\left(x \notin \mathbb{D}_i\right)\left(\mathbb{1}\left(h(x) \neq h_i(x)\right) - \left(\mathbb{1}\left(h(x) \neq y\right) - \mathbb{1}\left(h_i(x) \neq y\right)\right)\right)\right].$$

The desired result then follows from the inequality that

$$\mathbb{1}\left(h(x) \neq y\right) - \mathbb{1}\left(h_i(x) \neq y\right) \leq \mathbb{1}\left(h(x) \neq h_i(x)\right).$$

$\square$

**Lemma 9** (Deviation Bounds). *Pick* $0 < \delta < 1/e$. *With probability at least* $1 - \delta$ *the following holds. For all* $(h, h') \in \mathbb{H}^2$ *and* $\forall n \geq 3,$

$$|\widetilde{\mathrm{reg}}_n(h, h') - \mathrm{reg}(h, h', Z_n^\dagger)| \leq \sqrt{\eta_n(\widetilde{\mathrm{err}}_n(h) + \widetilde{\mathrm{err}}_n(h'))} + \eta_n \tag{19}$$

$$|\mathrm{err}_{\mathrm{dis}}(h, Z_n^\dagger) - \overline{\mathrm{err}}_n(h)| \leq \sqrt{\eta_n \overline{\mathrm{err}}_n(h)} + \eta_n, \tag{20}$$

*where*

$$\mathrm{err}_{\mathrm{dis}}(h, Z_n^\dagger) := \frac{1}{n} \sum_{i=1}^{n} \mathbb{1}\left(x_i \in \mathbb{D}_i\right) \mathbb{1}\left(h(x_i) \neq y_i\right),$$

$$\eta_n := \frac{12}{n} \ln\left(\frac{32n|\mathbb{H}| \ln n}{\delta}\right).$$

*Proof.* Our proof strategy is the following. Starting with a fixed $i$ and some fixed $(h, h') \in \mathbb{H}^2$, we apply the concentration result given by Lemma 3 of Kakade and Tewari [2009] to bound the deviations of the regret and error terms. Then we apply a union bound over $i$ and pairs of hypotheses to obtain the desired result. First consider the empirical regret. Define

$$
\begin{aligned}
R_i \quad \triangleq \quad & \mathbb{1}\,(x_i \in \mathbb{D}_i)\,(\mathbb{1}\,(h(x_i) \neq y_i) - \mathbb{1}\,(h'(x_i) \neq y_i)) + \\
& \mathbb{1}\,(x_i \notin \mathbb{D}_i)\,(\mathbb{1}\,(h(x_i) \neq h_i(x_i)) - \mathbb{1}\,(h'(x_i) \neq h_i(x_i))).
\end{aligned} \tag{21}
$$

Because $R_i$ is measurable with respect to $\mathcal{F}_i = \sigma(\{(x_j, y_j, u_j)\}_{j=1}^i)$, we have that $R_i - \mathbf{E}_i\,[R_i]$ is a martingale difference sequence adapted to the filtration $\mathcal{F}_i$. We also have $\mathbf{E}_i\,[R_i - \mathbf{E}_i\,[R_i]] \leq 2$ and

$$
\begin{aligned}
\mathbf{E}_i\,\left[(R_i - \mathbf{E}_i\,[R_i])^2\right] \quad &\leq \quad \mathbf{E}_i\,\left[R_i^2\right] & (22) \\
&= \quad \mathbf{E}_i\,\left[\mathbb{1}\,(x_i \in \mathbb{D}_i)\,(\mathbb{1}\,(h(x_i) \neq y_i) - \mathbb{1}\,(h'(x_i) \neq y_i))^2\right] + \\
& \quad\quad \mathbf{E}_i\,\left[\mathbb{1}\,(x_i \notin \mathbb{D}_i)\,(\mathbb{1}\,(h(x_i) \neq h_i(x_i)) - \mathbb{1}\,(h'(x_i) \neq h_i(x_i)))^2\right] & (23) \\
&\leq \quad \mathbf{E}_i\,\left[\mathbb{1}\,(x_i \in \mathbb{D}_i)\,(\mathbb{1}\,(h(x_i) \neq y_i) + \mathbb{1}\,(h'(x_i) \neq y_i))\right] + \\
& \quad\quad \mathbf{E}_i\,\left[\mathbb{1}\,(x_i \notin \mathbb{D}_i)\,(\mathbb{1}\,(h(x_i) \neq h_i(x_i)) + \mathbb{1}\,(h'(x_i) \neq h_i(x_i)))\right] & (24) \\
&= \quad \mathrm{err}_i^\dagger(h) + \mathrm{err}_i^\dagger(h'). & (25)
\end{aligned}
$$

Applying Lemma 3 of Kakade and Tewari [2009] to the sequence $R_i - \mathbf{E}_i\,[R_i]$, we have for any $i \geq 3$ and $0 < \delta_i < 1/e$, the following holds with probability at most $8 \ln(i)\delta_i$:

$$
|\mathrm{reg}(h, h', Z_i^\dagger) - \widetilde{\mathrm{reg}}_i(h, h')| \geq 2\sqrt{\frac{1}{i}(\widetilde{\mathrm{err}}_i(h) + \widetilde{\mathrm{err}}_i(h')) \ln(1/\delta_i)} + 6\ln(1/\delta_i)/i.
$$

Now consider the error terms. Let

$$
E_i := \mathbb{1}\,(x_i \in \mathbb{D}_i)\,\mathbb{1}\,(h(x_i) \neq y_i).
$$

Again, $E_i$ is measurable with respect to $\mathcal{F}_i$, and we have $|E_i - \mathbf{E}_i\,[E_i]| \leq 1$ and

$$
\mathbf{E}_i\,\left[(E_i - \mathbf{E}_i\,[E_i])^2\right] \leq \mathbf{E}_i\,\left[E_i^2\right] \leq \mathbf{E}_i\,[E_i].
$$

By using the same concentration lemma of Kakade and Tewari [2009] to the martingale difference sequence $E_i - \mathbf{E}_i\,[E_i]$, we have that the following holds

$$
|\,\mathrm{err}_{\mathrm{dis}}(h, Z_i^\dagger) - \overline{\mathrm{err}}_i(h)| \geq 2\sqrt{\frac{\overline{\mathrm{err}}_i(h)}{i}\ln(1/\delta_i)} + 3\ln(1/\delta_i)/i
$$

with probability at most $8 \ln(i)\delta_i$ for any $i \geq 3$, $0 < \delta_i < 1/e$ and $h$.

To bound the probability of the union of these large-deviation events over all $n \geq 3$ and all hypotheses, it suffices to choose $\delta_n = \delta/(n^2 32|\mathbb{H}|^2 \ln n)$, which leads to the desired result. $\square$

Using these two lemmas, we obtain the main theorem providing a generalization guarantee. In fact, here we prove a stronger result than Theorem 5, where the expected regret bound $\Delta_i^*$ is defined in terms of $\overline{\mathrm{err}}_i(h^*) \leq \mathrm{err}(h^*)$.

**Theorem 10** (Generalization Guarantee). *Pick any $0 < \delta < 1/e$ and let*

$$
\Delta_i^* := c_1\sqrt{\eta_i \overline{\mathrm{err}}_i(h^*)} + c_2(\eta_i + b_i).
$$

*With probability at least $1 - \delta$, the follow holds for all $i \geq 1$,*

$$
\begin{aligned}
\mathrm{reg}(h, h^*) \quad &\leq \quad 4\Delta_i^* \quad \textit{for all } h \in \mathbb{V}_{i+1}, \quad \textit{and} & (26) \\
\mathrm{reg}(h^*, h_{i+1}, Z_i) \quad &\leq \quad \Delta_i. & (27)
\end{aligned}
$$

*Proof.* Conditioning on the high probability event in Lemma 9, we prove this theorem by induction. For $i \leq 3$ both statements are true by the fact that regrets are upper-bounded by $1 \leq \min(\Delta_i, \Delta_i^*)$

for $i \leq 3$. Suppose the inductive hypothesis holds for $1 \leq i \leq m - 1$. We first prove (26) for $i = m$. Using the bound (19) from Lemma 9, we have for any $h \in \mathbb{V}_{m+1}$

$$\widetilde{\mathrm{reg}_m}(h, h^*) \tag{28}$$
$$\leq \mathrm{reg}(h, h^*, Z_m^\dagger) + \sqrt{\eta_m(\widetilde{\mathrm{err}}_m(h) + \widetilde{\mathrm{err}}_m(h^*))} + \eta_m$$
$$\leq \mathrm{reg}(h, h^*, Z_m) + b_m + \sqrt{\eta_m(\widetilde{\mathrm{err}}_m(h) + \widetilde{\mathrm{err}}_m(h^*))} + \eta_m$$
$$\leq \mathrm{reg}(h, h_{m+1}, Z_m) + \sqrt{\eta_m(\widetilde{\mathrm{reg}_m}(h, h^*) + 2\widetilde{\mathrm{err}}_m(h^*))} + \eta_m + b_m$$
$$\leq \Delta_m + \sqrt{\eta_m(\widetilde{\mathrm{reg}_m}(h, h^*) + 2\widetilde{\mathrm{err}}_m(h^*))} + \eta_m + b_m$$
$$\leq \Delta_m + \frac{\widetilde{\mathrm{reg}_m}(h, h^*)}{2} + \sqrt{2\eta_m\overline{\mathrm{err}}_m(h^*)} + \frac{3\eta_m}{2} + b_m$$
$$\leq (c_1\sqrt{\eta_m\,\mathrm{err}(h^*, Z_m)} + c_2(\eta_m + b_m)) + \frac{\widetilde{\mathrm{reg}_m}(h, h^*)}{2} + \sqrt{2\eta_m\overline{\mathrm{err}}_m(h^*)} + \frac{3\eta_m}{2} + b_m$$
$$\leq (c_1\sqrt{\eta_m\,\mathrm{err}(h^*, Z_m^\dagger)} + c_2(\eta_m + b_m)) + \frac{\widetilde{\mathrm{reg}_m}(h, h^*)}{2} + \sqrt{2\eta_m\overline{\mathrm{err}}_m(h^*)} + \frac{3\eta_m}{2} + b_m. \tag{29}$$

In the above, the second inequality is by the bound (10). The third inequality is by the fact that $h_{m+1} = \arg\min_{h \in \mathbb{H}} \mathrm{err}(h, Z_m)$. The fourth inequality is due to $h \in \mathbb{V}_{m+1}$, so it has a small empirical regret against the current ERM $h_{m+1}$. The fifth inequality involves more reasoning. By the inductive hypothesis that (27) holds for $1 \leq i \leq m - 1$, we have $h^* \in \mathbb{V}_i$ for $1 \leq i \leq m$. This and Lemma 8 imply that

$$\widetilde{\mathrm{reg}_m}(h, h^*) \geq \mathrm{reg}(h, h^*) \geq 0. \tag{30}$$

We then apply the inequality $\sqrt{a+b} \leq \sqrt{a} + \sqrt{b}$ for $a, b \geq 0$ and Cauchy-Schwarz to obtain

$$\sqrt{\eta_m(\widetilde{\mathrm{reg}_m}(h, h^*) + 2\widetilde{\mathrm{err}}_m(h^*))} \leq \sqrt{\eta_m\widetilde{\mathrm{reg}_m}(h, h^*)} + \sqrt{2\widetilde{\mathrm{err}}_m(h^*)}$$
$$\leq \frac{\widetilde{\mathrm{reg}_m}(h, h^*)}{2} + \frac{\eta_m}{2} + \sqrt{2\widetilde{\mathrm{err}}_m(h^*)}.$$

The sixth inequality is by the definition of $\Delta_m$ and the fact that $\mathrm{err}(h_{m+1}, Z_m) \leq \mathrm{err}(h^*, Z_m)$. The final inequality is by the bound (9). Because $h^* \in \mathbb{V}_i$ for $1 \leq i \leq m$, $h^*$ agrees with $h_i$ on all predicted labels for $1 \leq i \leq m$, implying

$$\mathrm{err}(h^*, Z_m^\dagger) = \mathrm{err}_{\mathrm{dis}}(h^*, Z_m^\dagger). \tag{31}$$

Applying the deviation bound (20) and Cauchy-Schwarz, we get

$$\mathrm{err}_{\mathrm{dis}}(h^*, Z_m^\dagger) \leq \frac{3}{2}(\overline{\mathrm{err}}_m(h^*) + \eta_m). \tag{32}$$

Combining (29), (30), (31), and (32) we obtain

$$\mathrm{reg}(h, h^*) \leq \widetilde{\mathrm{reg}_m}(h, h^*)$$
$$\leq 2\left(c_1\sqrt{\frac{3}{2}\eta_m(\overline{\mathrm{err}}_m(h^*) + \eta_m)} + c_2(\eta_m + b_m)\right) + 2\sqrt{2\eta_m\overline{\mathrm{err}}_m(h^*)} + 3\eta_m + 2b_m$$
$$= (c_1\sqrt{6} + 2\sqrt{2})\sqrt{\eta_m\overline{\mathrm{err}}_m(h^*)} + (c_1\sqrt{6} + 2c_2 + 3)\eta_m + (2c_2 + 2)b_m$$
$$\leq 4\Delta_m^*,$$

where the last inequality is by our choices of $c_1$ and $c_2$. We thus establish (26) for $i = m$.

Next we prove (27) for $i = m$. Again, starting with the deviation bound (19) we have

$$\mathrm{reg}(h^*, h_{m+1}, Z_m^\dagger) \leq \widetilde{\mathrm{reg}_m}(h^*, h_{m+1}) + \sqrt{\eta_m(\widetilde{\mathrm{err}}_m(h^*) + \widetilde{\mathrm{err}}_m(h_{m+1}))} + \eta_m$$
$$= \widetilde{\mathrm{reg}_m}(h^*, h_{m+1}) + \sqrt{\eta_m(2\widetilde{\mathrm{err}}_m(h^*) + \widetilde{\mathrm{reg}_m}(h_{m+1}, h^*))} + \eta_m.$$

As explained earlier, we have $h^* \in \mathbb{V}_i$ for $1 \leq i \leq m$ by the inductive hypothesis (27), which implies that $\widetilde{\mathrm{err}}_m(h^*) = \overline{\mathrm{err}}_m(h^*)$ and $\widetilde{\mathrm{reg}_m}(h_{m+1}, h^*) \geq \mathrm{reg}(h_{m+1}, h^*) \geq 0$ (by Lemma 8). Thus we

have

$$
\begin{aligned}
\mathrm{reg}(h^*, h_{m+1}, Z_m^\dagger) &\leq \widetilde{\mathrm{reg}}_m(h^*, h_{m+1}) + \frac{1}{2}\widetilde{\mathrm{reg}}_m(h_{m+1}, h^*) + \sqrt{2\eta_m \overline{\mathrm{err}}_m(h^*)} + \frac{3}{2}\eta_m \\
&= \frac{1}{2}\widetilde{\mathrm{reg}}_m(h^*, h_{m+1}) + \sqrt{2\eta_m \overline{\mathrm{err}}_m(h^*)} + \frac{3}{2}\eta_m \\
&\leq \sqrt{2\eta_m \overline{\mathrm{err}}_m(h^*)} + \frac{3}{2}\eta_m.
\end{aligned}
\tag{33}
$$

The deviation bound (20) implies that

$$
\overline{\mathrm{err}}_m(h^*) \leq 2\,\mathrm{err}_{\mathrm{dis}}(h^*, Z_m^\dagger) + 3\eta_m = 2\,\mathrm{err}(h^*, Z_m^\dagger) + 3\eta_m,
\tag{34}
$$

where the equality is due to $h^* \in \mathbb{V}_i$ for $1 \leq i \leq m$. Combining (33) and (34), we get

$$
\begin{aligned}
\mathrm{reg}(h^*, h_{m+1}, Z_m^\dagger) &\leq \sqrt{2\eta_m(2\,\mathrm{err}(h^*, Z_m^\dagger) + 3\eta_m)} + \frac{3}{2}\eta_m \\
&\leq 2\sqrt{\eta_m(\mathrm{err}(h^*, Z_m) + b_m)} + (\sqrt{6} + 3/2)\eta_m \\
&= 2\sqrt{\eta_m(\mathrm{reg}(h^*, h_{m+1}, Z_m) + \mathrm{err}(h_{m+1}, Z_m)) + \eta_m b_m} + (\sqrt{6} + 3/2)\eta_m \\
&\leq \frac{1}{2}\mathrm{reg}(h^*, h_{m+1}, Z_m) + 2\sqrt{\eta_m\,\mathrm{err}(h_{m+1}, Z_m)} + 2\sqrt{\eta_m b_m} + (\sqrt{6} + 7/2)\eta_m \\
&\leq \frac{1}{2}\mathrm{reg}(h^*, h_{m+1}, Z_m) + 2\sqrt{\eta_m\,\mathrm{err}(h_{m+1}, Z_m)} + (\sqrt{6} + 9/2)\eta_m + b_m.
\end{aligned}
$$

This and the bound (10) imply that

$$
\mathrm{reg}(h^*, h_{m+1}, Z_m) \leq 4\sqrt{\eta_m\,\mathrm{err}(h_{m+1}, Z_m)} + (2\sqrt{6} + 9)\eta_m + 4b_m \leq \Delta_m.
$$

$\square$

Next we provide a proof for the query complexity bound. Again, we prove a stronger result that uses $\overline{\mathrm{err}}_n(h^*)$ in place of $\mathrm{err}(h^*)$ in Theorem 6.

**Theorem 11** (Query Complexity Bound). *Under the conditions of Theorem 10, with probability at least $1 - \delta$ the following holds: $\forall n > 0$, $Q_n$ is bounded by*

$$
4\theta\,\mathrm{err}(h^*)n + \theta \cdot \mathrm{O}\left(\sqrt{n\overline{\mathrm{err}}_n(h^*)\ln(n|\mathbb{H}|/\delta)\ln^2 n} + \ln(n|\mathbb{H}|/\delta)\ln n + nb_n \ln n) + 8\ln(8n^2 \ln n/\delta)\right).
$$

*Proof.* The random variable $\mathbb{1}(x_i \in \mathbb{D}_i)$ is measurable with respect to $\mathcal{F}_i := \sigma(\{(x_j, y_j, u_j)\}_{j=1}^i)$, so

$$
R_i := \mathbb{1}(x_i \in \mathbb{D}_i) - \mathbf{E}_i[\mathbb{1}(x_i \in \mathbb{D}_i)]
$$

forms a martingale difference sequence adapted to the filtration $\mathcal{F}_i, i \geq 1$. Moreover, we have $|R_i| \leq 1$ and

$$
\mathbf{E}_i[R_i^2] \leq \mathbf{E}_i[\mathbb{1}(x_i \in \mathbb{D}_i)].
$$

Applying Lemma 3 of Kakade and Tewari [2009] with the above bounds and Cauchy-Schwarz, we get that with probability at least $1 - \delta$,

$$
\forall n \geq 3, \quad Q_n \leq 2\sum_{i=1}^n \mathbf{E}_i[\mathbb{1}(x_i \in \mathbb{D}_i)] + 8\ln(4n^2(\ln n)/\delta).
\tag{35}
$$

We next bound the sum of the conditional expectations. Pick some $i$ and consider the case $x_i \in \mathbb{D}_i$. Define

$$
\bar{h} := \begin{cases} h_i, & h_i(x_i) \neq h^*(x_i), \\ h', & h'(x_i) \neq h^*(x_i), \end{cases}
$$

where

$$
\begin{aligned}
h_i &:= \arg\min_{h \in \mathbb{H}} \mathrm{err}(h, Z_{i-1}), \\
h' &:= \arg\min_{h \in \mathbb{H} \wedge h(x_i) \neq h_i(x_i)} \mathrm{err}(h, Z_{i-1}).
\end{aligned}
\tag{36}
\tag{37}
$$

Because $x_i \in \mathbb{D}_i$, we have $h' \in \mathbb{V}_i$, implying $\bar{h} \in \mathbb{V}_i$. Conditioned on the high probability event in Theorem 10, we have $h^* \in \mathbb{V}_i$ and hence

$$
\begin{aligned}
\mathbf{E}_{x \sim \mu_{\mathbb{X}}} \left[ \mathbb{1} \left( \bar{h}(x) \neq h^*(x) \right) \right] &= \mathbf{E}_{x \sim \mu_{\mathbb{X}}} \left[ \mathbb{1} \left( \bar{h}(x) \neq h^*(x) \wedge x \in \mathbb{D}_i \right) \right] \\
&\leq \operatorname{err}_i(\bar{h}) + \operatorname{err}_i(h^*) \\
&= \operatorname{reg}_i(\bar{h}) + 2\operatorname{err}_i(h^*) \\
&\leq 4\Delta_{i-1}^* + 2\operatorname{err}_i(h^*),
\end{aligned}
$$

where the last inequality is by Theorem 10 and the condition that both $\bar{h}$ and $h^*$ are in $\mathbb{V}_i$. This implies that

$$
x_i \in \mathbb{D}(\{h \mid \mathbf{E}_{x \sim \mu_{\mathbb{X}}} \left[ \mathbb{1} \left( h(x) \neq h^*(x) \right) \right] \leq 4\Delta_{i-1}^* + 2\operatorname{err}_i(h^*)\}).
$$

We thus have

$$
\begin{aligned}
\mathbf{E}_i \left[ \mathbb{1} \left( x_i \in \mathbb{D}_i \right) \right] &\leq \mathbf{E}_i \left[ \mathbb{1} \left( x_i \in \mathbb{D}(\{h \mid \mathbf{E}_{x \sim \mu_{\mathbb{X}}} \left[ \mathbb{1} \left( h(x) \neq h^*(x) \right) \right] \leq 4\Delta_{i-1}^* + 2\operatorname{err}_i(h^*)\})) \right] \\
&\leq \theta(4\Delta_{i-1}^* + 2\operatorname{err}_i(h^*)),
\end{aligned} \tag{38}
$$

where the last inequality uses the definition of the disagreement coefficient:

$$
\theta = \theta(h^*) \triangleq \sup_{r>0} \frac{\mathbf{E}_{x \sim \mu_{\mathbb{X}}} \left[ \mathbb{1} \left( \exists h \in \mathbb{H} \, \text{s.t.} \, h^*(x) \neq h(x), \, \mathbf{E}_{x' \sim \mu_{\mathbb{X}}} \left[ \mathbb{1} \left( h(x') \neq h^*(x') \right) \right] \leq r \right) \right]}{r}. \tag{39}
$$

Summing (38) over $i \in \{1, \ldots, n\}$ and noting that the high probability event in Theorem 10 holds over all rounds, we get that with probability at least $1 - \delta$,

$$
\forall n \geq 3, \quad \sum_{i=1}^{n} \mathbf{E}_i \left[ \mathbb{1} \left( x_i \in \mathbb{D}_i \right) \right] \leq 3 + \sum_{i=4}^{n} \theta(4\Delta_{i-1}^* + 2\operatorname{err}_i(h^*)) \tag{40}
$$

$$
\leq 3 + 2n\theta\overline{\operatorname{err}}_n(h^*) + 4\theta \sum_{i=4}^{n} \Delta_{i-1}^* \tag{41}
$$

$$
= 3 + 2n\theta\overline{\operatorname{err}}_n(h^*) + 4\theta \sum_{i=4}^{n} \frac{1}{i-1}(i-1)\Delta_{i-1}^*. \tag{42}
$$

For all $i \leq n$, we have

$$
i\Delta_i^* = c_1 \sqrt{i^2 \eta_i \overline{\operatorname{err}}_i(h^*)} + ic_2(\eta_i + b_i) \tag{43}
$$

$$
\leq c_1 \sqrt{n^2 \eta_n \overline{\operatorname{err}}_n(h^*)} + nc_2(\eta_n + b_n) \tag{44}
$$

$$
= n\Delta_n^* \tag{45}
$$

by plugging in the definitions of $\eta_i$ and $\overline{\operatorname{err}}_i(h^*)$. Therefore, we have

$$
\sum_{i=1}^{n} \mathbf{E}_i \left[ \mathbb{1} \left( x_i \in \mathbb{D}_i \right) \right] \leq 3 + 2n\theta\overline{\operatorname{err}}_n(h^*) + 8\theta n \Delta_n^* \ln(n) \tag{46}
$$

$$
= 3 + 2n\theta\overline{\operatorname{err}}_n(h^*) \tag{47}
$$

$$
+ \theta \mathrm{O} \left( \sqrt{n\overline{\operatorname{err}}_n(h^*) \left( \ln \left( \frac{n|\mathbb{H}|}{\delta} \right) \ln^2 n \right)} + \ln \left( \frac{n|\mathbb{H}|}{\delta} \right) \ln n + nb_n \ln n \right).
$$

Combining this and (35) via a union bound leads to the desired result.

## D  Specialization to Oracle Epiphany

Here we prove Corollary 7. First, Lemma 2 and a union bound over the $K$ unknown regions show that for any fixed $t > 0$,

$$
\Pr \left\{ tb_t \leq \frac{K}{\beta} \ln \frac{4K}{\delta} \right\} \geq 1 - \delta/4. \tag{48}
$$

Conditioning on the high-probability events in (48) and Theorem 10, we will find $t$ such that

$$
4\Delta_t^* \leq C \cdot \left( \sqrt{\frac{e^* \ln(t|\mathbb{H}|/\delta)}{t}} + \frac{\ln(t|\mathbb{H}|/\delta)}{t} + \frac{\widetilde{K}}{\beta t} \right) \leq \epsilon, \tag{49}
$$

where $C$ is some absolute constant. We do this in two steps. First, we find $t_1$, $t_2$ and $t_3$ that satisfy

$$\epsilon \geq C \cdot \sqrt{\frac{e^* \ln(t_1|\mathbb{H}|/\delta)}{t_1}}, \qquad \epsilon \geq C \cdot \frac{\ln(t_2|\mathbb{H}|/\delta)}{t_2} \quad \text{and} \quad \epsilon \geq C \cdot \frac{\widetilde{K}}{\beta t_3}$$

respectively. This gives

$$t_1 = \mathrm{O}\left(\frac{e^*}{\epsilon^2} \ln \frac{|\mathbb{H}|}{\epsilon^2 \delta}\right), \qquad t_2 = \mathrm{O}\left(\frac{1}{\epsilon} \ln \frac{|\mathbb{H}|}{\epsilon \delta}\right), \quad \text{and} \quad t_3 = \mathrm{O}\left(\frac{\widetilde{K}}{\beta \epsilon}\right).$$

Setting $t_\epsilon = t_1 + t_2 + t_3$ then gives the desired form for $t_\epsilon$. To bound the query complexity, we substitute $t_\epsilon$ for $n$ in the query complexity bounds (47) and (35), and use that fact that $4\Delta_{t_\epsilon}^* \leq \epsilon$ and $\overline{\mathrm{err}}_{t_\epsilon}(h^*) \leq \mathrm{err}(h^*)$. Thus we obtain

$$Q_{t_\epsilon} \leq 3 + \theta \cdot \left(2t_\epsilon e^* + 2t_\epsilon \epsilon \ln t_\epsilon + 8\ln(4t_\epsilon^2 \ln(t_\epsilon)/\delta)\right). \tag{50}$$

Plugging $t_\epsilon$ into the expression above leads to the desired query complexity bound. $\qquad\square$

# E   Details for OCR Experiments

We implement an approximate version of Alg. 2 that uses *online optimization*. This implementation is based on online logistic regression in Vowpal Wabbit (`hunch.net/~vw`). It processes the data in one pass, updating an approximate ERM and performing an online disagreement test with a reverting weight technique [Karampatziakis and Langford, 2011, Appendix F]. This online test costs $\mathrm{O}(d)$ time per new example, where $d$ is the average number of features. While being efficient in practice, this online algorithm may not retain the theoretical guarantees of Alg. 2.

We obtain the MNIST data from the LIBSVM dataset page[3]. The training and testing sets have 60,000 and 10,000 examples, respectively. Table 1 shows the percentages of the ten digits in the data. We use online linear logistic regression in Vowpal Wabbit[4] (VW) as our base supervised learning

Table 1: Percentages of ten digits in MNIST

| digit | 0 | 1 | 2 | 3 | 4 | 5 | 6 | 7 | 8 | 9 |
|---|---|---|---|---|---|---|---|---|---|---|
| train (%) | 9.87 | 11.24 | 9.93 | 10.22 | 9.74 | 9.04 | 9.86 | 10.44 | 9.75 | 9.92 |
| test (%) | 9.80 | 11.35 | 10.32 | 10.10 | 9.82 | 8.92 | 9.58 | 10.28 | 9.74 | 10.09 |

algorithm, with the default learning rate and bit precision. We consider the binary classification task of "5" vs. other digits, and pick $\mathbb{U}$ based on the binary classification error for "5" vs. every other single digit, summarized in Table 2. In addition to single digits, we also consider multiple digits

Table 2: Binary classification error for "5" vs. every other digit

| digit | 0 | 1 | 2 | 3 | 4 | 6 | 7 | 8 | 9 |
|---|---|---|---|---|---|---|---|---|---|
| error (%) | 1.1 | 0.4 | 2.3 | 4.4 | 0.8 | 2.5 | 0.6 | 4.6 | 1.3 |

as $\mathbb{U}$. In particular, we start from both ends of the confusion spectrum and include more digits into $\mathbb{U}$. This results in a total of six settings of $\mathbb{U}$: $\{"8"\}$, $\{"8", "3"\}$, $\{"8", "3", "6"\}$, $\{"1"\}$, $\{"1", "7"\}$ and $\{"1", "7", "4"\}$. Fig. 3 shows the median, the 25th and 75th quantiles of the smallest number of queries for achieving a test error of 4% over 100 random trials, for the six different $\mathbb{U}$'s. For the less confusing $\mathbb{U}$'s, Oracular-EPICAL always performs much better than passive when $\mathbb{U} = \{"1"\}$, but starts approaching passive as $\mathbb{U}$ gets larger and $\beta$ gets smaller. For the more confusing $\mathbb{U}$'s, it is interesting that $\beta$ has a much weaker effect for $\mathbb{U} = \{"8"\}$ than for $\mathbb{U} = \{"3"\}$ (see Section 5), which are almost equally confused with "5". One possibility is that they are confused with "5" in different sub-spaces of the feature space, and the confusion with "8" could somehow be resolved by learning from other digits, while the confusion with "3" cannot. The size of $\mathbb{U}$ does not have a

`{mnist.bz2, mnist.t.bz2}`
[4]`hunch.net/~vw`

(a) $\mathbb{U} = \{$"8"$\}$

(b) $\mathbb{U} = \{$"8", "3"$\}$

(c) $\mathbb{U} = \{$"8", "3", "6"$\}$

(d) $\mathbb{U} = \{$"1"$\}$

(e) $\mathbb{U} = \{$"1", "7"$\}$

(f) $\mathbb{U} = \{$"1", "7", "4"$\}$

Figure 3: Oracular-EPICAL results on MNIST (median, 25th and 75th quantiles)

clear effect because the steep increase in the number of queries over decreasing $\beta$ could very well be caused by $\mathbb{U} = \{$"3"$\}$.

In addition to the average performance demonstrated so far, we are also interested in the tail performance, which better aligns with our high-probability bounds. Fig. 4 and Fig. 5 show the 95th and 85th quantiles of number of queries over 100 random trials, respectively. Oracular-EPICAL performs surprisingly poor for $\beta = 1$ across all $\mathbb{U}$'s at the 95th quantile. Further investigation shows that in roughly 15% of the random trials for $\beta = 1$, Oracular-EPICAL becomes overly confident about its own (mis-)predicted labels, stops querying prematurely, and never recovers from that bias. This is caused by an overly small "mellowness" parameter, which is a tuning parameter in our online implementation of Oracular-EPICAL that controls the multiplicative constant in the threshold $\Delta_t$. A larger mellowness parameter improves the tail performance for $\beta = 1$, but reduces the average improvement over passive learning across all $\beta$. Thus, choosing a proper mellowness parameter in a data-dependent, active learning setting is an important practical issue for further investigation.

Figure 4: Oracular-EPICAL results on MNIST (95th quantile)

Figure 5: Oracular-EPICAL results on MNIST (85th quantile)