[Reviews · NeurIPS 2016]

Reviewer 1

Summary

This paper proposes a new model for active learning, so-called active learning with oracle epiphany, which purports to describe realistic oracles (e.g., human annotators) that may fail to correctly classify certain examples until a certain point when enough of such examples had been presented (this is the moment of oracle's epiphany). In mathematical terms, this is accomplished by splitting the instance space into two disjoint subsets, on one of which the oracle knows the label, while the other is further split into a finite number of disjoint subsets that belong to different categories. Before the oracle had an epiphany, whenever it encounters an instance from one of these sets, it emits the abstention (don't-know) symbol, but, with a certain fixed probability, the next time it sees an instance from that subclass, it may have an epiphany and start labeling instances from that subclass correctly from that moment on. The authors present two algorithms that build on the classical active learning scheme of Cohn, Atlas, and Ladner (CAL) to work with oracle epiphany: EPICAL and Oracular-EPICAL, and present high-probability upper bounds on the total number of queries to achieve a given accuracy with a given confidence. The bounds involve the disagreement coefficient (as expected) and the VC dimension of the hypothesis class (again, as expected), plus a contribution that depends on the probability of oracular epiphany and on the number of subclasses where the oracle may experience an epiphany. The theoretical results are supplemented with empirical evaluation on synthetic and real data.

Qualitative Assessment

This paper is a nice addition to the literature on active learning, straddling the theoretical realm and the practically motivated algorithm design issues that arise when dealing with realistic oracles (e.g., human annotators who may initially not have enough confidence to generate labels for the given queries, but would gain confidence after having seen enough queries of this sort). Granted, the theoretical model is stripped down and simple, but this is a good place to start from. My only quibble is the absence of lower bounds, which may be too much to ask for, but it seems that the theoretical model is clean enough to at least attempt to derive one.

Confidence in this Review

3-Expert (read the paper in detail, know the area, quite certain of my opinion)


Reviewer 2

Summary

This work studies the problem of active learning in a setting where the oracle might not be sure how to answer queries in certain (unknown) regions of the space until asked a few times for samples from those regions. The problem is well-motivated by behavioral studies of human labelers, and by intuitive descriptions of cases that might be seemingly ambiguous at first (e.g., when asked whether a webpage is about sports, the labeler might initially not make up her mind about how to label webpages about sports memorabilia). The paper models this phenomenon by supposing there are K unknown regions of the space, and each time a query is made in one of these regions for which an "epiphany" has not yet occurred, there is a beta probability the epiphany occurs then, in which case the oracle returns a label as usual (and will henceforth return a label whenever points in that region are queried), and otherwise the oracle abstains from labeling. They propose simple modifications of known active learning algorithms (CAL in the realizable case, OracularCAL in the agnostic case), and analyze the query complexity in each case, paying particular attention to the increase in query complexity induced by these abstentions, which is generally a function of K and beta.

Qualitative Assessment

Overall I enjoyed reading this paper. It is very well written, and the algorithms and analysis seem to be natural modifications of these existing approaches to active learning. The theoretical issues that arise in handling these abstentions and quantifying their effect on the query complexity are at times nontrivial, and are handled in elegant and appropriate ways. I suspect that not many people are aware of this problem, but it is quite well motivated in the paper, and seems to be a good problem to study. The specific theoretical model proposed for this phenomenon is, however, a bit toy-like. From the motivation, it seems these epiphanies might have more to do with the labeler needing enough data to get a feel for what the distribution of samples will be, to see where to draw the line, rather than there being some random internal event that could happen at any time upon being queried for a data point of that type. But I suppose this simplistic model could at least be a reasonable starting place, which can hopefully be made more realistic or general in future work. I have a few technical comments for the authors: The halting criterion for EPICAL is that \mu_X(D) \le \epsilon. However, it seems far preferable to halt upon \sup_{h,h' \in V} \mu_X(h(x)\neq h'(x)) \le \epsilon. Not only is this more closely related to the error rate, but it would guarantee that the algorithm actually halts (with query complexity at most roughly the passive sample complexity); in contrast, the halting criterion as currently stated might never actually be satisfied (e.g., interval classifiers with target as empty interval), so that the algorithm never halts. In Theorem 1, it seems clear that the terms M_CAL and 1/beta are unavoidable (as argued on page 4). It would be nice to also have some example(s) illustrating the kinds of scenarios where the \bar{M} term is also unavoidable and nonredundant. If the region U has large probability, then we'll very quickly get 1/beta abstentions, so \bar{M} seems unnecessary. But if U has small probability (eg, \epsilon/2), then we don't care about the abstentions anyway. So it seems the medium-sized U case is where \bar{M} might be needed in the bound. Is there a good example to illustrate its necessity in bounding the query complexity of EPICAL? In Corollary 7, in the query complexity bound, the first appearance of \tilde{d} in this expression is redundant (the second term already includes a value greater then \tilde{d}(e*/\epsilon) toward the right-most side of the expression). In the paragraph after Corollary 7, it is claimed that the leading term in the agnostic setting is of order \theta (e*/\epsilon)(\tilde{d}+\tilde{K}/\beta). However, the bound also includes a term of order \theta (e*/\epsilon)^2 \tilde{d}. So the correct expression here would be \theta ( (e*/\epsilon)^2 \tilde{d} + (e*/\epsilon)\tilde{K}/\beta ).

Confidence in this Review

3-Expert (read the paper in detail, know the area, quite certain of my opinion)


Reviewer 3

Summary

The paper introduces a new active learning model, which attempts to incorporate previous empirical studies on human behavior in answering the label queries of learning algorithms. It is assumed that there is a partitioned subset of the input space where the response to a query is ``don’t know’’ until an ``epiphany’’ happens and from then on the correct answer is given for that subset. The paper considers natural extensions of previous active learning algorithms to this setting, and proves query complexity bounds. The bounds essentially contain extra additive factors, which depend linearly on the number of classes in the partition, and inversely on the probability of epiphany.

Qualitative Assessment

The paper deals with an important problem: to bring active learning closer to practical applications. The discussion of previous literature on the topic should mention work on the same problem for membership queries in query learning. The paper’s main source is the previous Oracular-CAL algorithm; however, the paper by Huang et al.(2015) is mainly on the algorithm ACTIVE-COVER; the relationship of this and Hsu (2010) should be clarified. The main feature of human oracles is making errors. The role of noise in active learning is, therefore the first issue to discuss in the present context. This seems to be missing from the paper and it should be added. As a possible extension of the model, reversibility of epiphanies is mentioned at the end; this is one (perhaps not the most natural) form of the imperfectness of human oracles, and the issue should be discussed in more detail. The paper mentions the ``unavoidable’’ K/beta cost in complexity; it would be useful to add some comments on the possibility of proving this unavoidability. The term ``unique’’ in Section 2 could perhaps be replaced by ``unseen’’. It is mentioned that for continuous distributions this assumption is without loss of generality. A comment should be added on other cases. The proof of Corollary 7 (the main result) from Theorem 6 is short; and at least some part of it (explaining how Lemma 2 connects the previous result to the result to be proven) would be illuminating to the readers, so it could perhaps be squeezed into the text. The paper makes a reasonable first step on an important problem. It is of good technical quality, making competent use of previous work in the standard model of active learning.

Confidence in this Review

2-Confident (read it all; understood it all reasonably well)


Reviewer 4

Summary

The paper provide theoretical analysis of active learning with oracles abstaining on difficult queries until accumulating enough information to make decisions. The analysis shows that active learning is possible with oracle epiphany, but incurs an additional cost depending on when epiphany happens.

Qualitative Assessment

The paper consider an interesting setting for machine learning, which may be of great interests to the Active learning literature. However, in my opinion, its theoretical results and the techniques used in the paper are expected. As seen in the paper, the analyses of the new algorithms are not much different from the standard settings, only that we need to account for the additional cost of waiting for the epiphany to happen, which appear to be not difficult to predict and quantify. In my opinion, the contributions in both theoretical techniques and algorithmic ideas of the paper are quite minimal.

Confidence in this Review

2-Confident (read it all; understood it all reasonably well)


Reviewer 5

Summary

The author considers a more realistic oracle for active learning which is called oracle epiphany. Under this new oracle, the authors analyze the query complexity for both the realizable and agnostic case.

Qualitative Assessment

The paper is well written. It starts from realizable case to get a good intuition about the query complexity then move on to the agnostic case. However I'm not familiar with active learning at all.

Confidence in this Review

1-Less confident (might not have understood significant parts)


Reviewer 6

Summary

The paper studies a new type of oracle in active learning, that has "epiphanies", modeling the setting that the oracle may initially answer some 'Don't know' if the query x is in some "uncertain" region (and end up answering at some point). The epiphany is modeled as geometric trials. Both realizable and agnostic cases are considered, and algorithms similar to CAL/Agnostic CAL are analyzed. Simulations show the effect of epiphany region (U) / epiphany probability (beta) to the label complexity of the algorithms. Specifically, if the epiphany region is important, then it is important to have a high epiphany probability (for active learning to outperform passive learning), and vice versa.

Qualitative Assessment

Technical Quality: the problem setting and the approach used in the paper are quite sound. For Theorem 1, the query complexity is a bit obscure, since it involves the disagreement coefficient restricted to the "known" region K, and it is not clear how to relate it with the global disagreement coefficient. Also, the stopping criterion is Algorithm 1 is a bit unsatisfying -- it is the size of the disagreement region rather than an upper confidence bound on the error that is measured (and compared against target error epsilon). Another direction worth investigating is how the disagreement region of the version space and unknown region in oracle epiphany interact with each other. If some theory can be done here, it will strongly support the experiments. Novelty and Originality: the "oracle with epiphany" model is new to me, and I think it is interesting. The proofs of Algorithm 2 is more-or-less standard (modulo the novel setting of rejection threshold involving b_t here.) Potential Impact: this paper studies a new oracle for active learning with is theoretically approachable, and it will also give inspirations on applied active learning research. Clarify and presentation: the paper is well written overall; the experiments and the theory fits well altogether. I find the proof of Lemma 4 a bit confusing -- in line 140, I think the high level idea is that "if we have used label budget >= \bar{m} + 2/beta\ln2/\delta, then we will definitely trigger epiphany". I suggest a bit revision on this paragraph. For Algorithm 2, I was once wondering if we can do the following modification: the algorithm does not keep counter b_t. At time step t, if the oracle returns \perp, the algorithm simply skips this iteration and pretend this example "does not exist". The examples (with hidden labels) collected this way would still be iid. This way, the label complexity of the algorithm seems to be at most the label complexity of Agnostic CAL + O(K/\beta), where the second term is the price of epiphany. Conceptually we can also think Algorithm 1 in this way as well. -- But it turns out that the iid property of the data is now violated, hence this modification does not work out -- perhaps the authors can remark on this?

Confidence in this Review

2-Confident (read it all; understood it all reasonably well)